# Effects of Ecological Programs and Other Factors on Soil Wind Erosion between 1981–2020

Jinzhou Wu [1,2,3,4,5] , Xiao Zheng [1,2,3,4,*], Lanlin Zhao [1,2,3,4,6], Junmei Fan [1,2,3,4,6] and Jinghong Liu [1,2,3,4,6]

1   CAS Key Laboratory of Forest and Management, Institute of Applied Ecology, Shenyang 110016, China
2   Qingyuan Forest, National Observation and Research Station, Shenyang 110016, China
3   Qingyuan Forest CERN, Chinese Academy of Sciences, Shenyang 110016, China
4   Key Laboratory for Management of Non-Commercial Forests, Shenyang 110016, China
5   Cartography and GIS Research Group, Department of Geography, Vrije Universiteit Brussel, 1050 Brussels, Belgium
6   University of Chinese Academy of Sciences, Beijing 100049, China
*   Correspondence: xiaozheng@iae.ac.cn

**Abstract:** Wind erosion is one of the most widespread and severe natural hazards in arid, semiarid, and semihumid regions worldwide. The Three-North region (TNR) (Northeast China, North China, and Northwest China) of China includes 90% of the wind erosion area in China. In response to the harsh environmental conditions in the TNR, China initiated a series of ecological programs, including the Three-North Afforestation Program and Grain for Green. However, little is known about the effect of these ecological programs on wind erosion. Therefore, within our study, we estimated the spatiotemporal variations in wind erosion in the TNR between 1981–2020 with a revised wind erosion model and analyzed its driving mechanism. Then, the ecological programs' effects on wind erosion changes was identified. The results showed the following. (1) From 1981 to 2020, wind erosion showed a clear downward trend of 99.02 t km$^{-2}$ a$^{-1}$, with a slope. On average, the areas of mild, moderate, severe, more severe, and very severe wind erosion accounted for 28.76%, 7.17%, 3.92%, 3.72%, and 13.29% of the total in the TNR, respectively. (2) Wind erosion variation was inconsistent in different parts of the TNR. The wind erosion expressed a long-term decreasing trend in Northeast China and the Loess Plateau, a nonsignificant change in North Central China, and an increasing trend in Northwest China. (3) On average, ecological programs were very limited in reducing erosion at the regional scale, with a contribution of approximately 5.93% in the TNR because of the relatively small scope of ecological programs' implementation. Climate change played a key role in adjusting wind erosion; wind speed, temperature, and precipitation affected 57.58% of the TNR. Human interference (proportion of cropland and grassland areas in a 1 km ×1 km grid) affected 8.78% of the TNR. Thus, the persistent complement of ecological programs, reasonable human activities, and timely observation is a method to alleviate wind erosion in the TNR.

**Keywords:** wind erosion; climate change; North China; ecological programs; human interference

## 1. Introduction

One of the most severe environmental and ecological issues facing arid and semiarid regions, soil wind erosion, affects more than 28% of the global landmass [1–4]. In natural ecosystems, wind erosion has caused land degradation and soil productivity reduction by depleting the number of fine particles and changing the structure of the nutrients and the composition of the soil [5,6]. Furthermore, in the human social system, the process of soil wind erosion has caused losses to public utilities [7]; for example, soil deposits impede road traffic, suspended dust reduces visibility, dust storms damage crop seedlings, and particle-related air pollution poses a health hazard [1,8].

China is among the countries most severely affected by wind erosion globally. Under the 5th Desertification and Sandification Monitoring and Survey in 2015, the aeolian

desertification area in China is 1.83 million km$^2$, occupying 69.93% of the total desertified land area nationwide [9], and 90% of soil wind erosion zones are concentrated in the Three-North region of China (TNR) (including Northeast China, North Central China, and Northwest China) [8]. With its widely distributed aeolian desert, low precipitation, frequent droughts, and high wind speeds, the TNR is prone to severe dust storms and has experienced severe wind erosion over a long time [10–12]. In response, and to improve the ecological environment, the government implemented a series of ecological programs, including the Three-North Afforestation Program (TANP) and Green for Grain [13,14]. However, the effect of ecological programs on the variation on soil wind erosion has not been scientifically and quantitatively assessed due to the lack of spatiation data from ecological programs [15,16].

Soil wind erosion can result from the interaction of natural factors, e.g., weather factors (wind speed, temperature, precipitation, and evaporation), soil texture, topography, and land cover [7,17]. To date, wind erosion assessment methods have varied significantly, from empirical to processed-based models and from qualified analysis models to quantified analysis models [18,19]. The Universal Wind Erosion Model (UWEM) was released by the USDA in 1961 and was later modified and released as the Wind Erosion Model (WEM) [20]; then, a revised UWEM was released (the RWEQ) [21], which is well-known and widely used [22]. Detailed climatic, soil, land cover, and land management data are required for these models. These models are typically applied on a field or small regional scale; however, they are generally not feasible for wind erosion analysis over large areas due to limitations in data accessibility [23,24]. Nevertheless, geographic information systems (GIS) and remote sensing (RS) have been adopted as powerful tools to assess wind erosion. Satellite imagery is particularly important in providing regular information about inaccessible areas [25].

Currently, several studies have been carried out on the wind erosion dynamics in the world's arid and semi-arid lands (e.g., Central Asia, Southern Africa, Algeria, Eastern Austria, East Africa and Europe) using the RWEQ [26–30]. The core of the research focuses on the impact of climate change on wind erosion and erosion risk assessment [27–29]. Fenta et al. [27] found that Sudan and Somalia have the severest wind erosion in East Africa. Borrelli et al. [28] try to map the pan European wind erosion mapping, and found that integrating wind erosion and environmental variables were a good idea to identified wind erosion. Scheper et al. [29] fulfilled the wind erosion risk mapping of Eastern Austria and found august was the month with the highest modeled soil loss. Li et al. [30] predict the wind erosion change trend of central Asia in the 21st century and found that the wind erosion of Northeastern Central Asia would increase while other area will decrease.

Since arid and semi-arid areas account for more than 42% of the national land area and are mainly concentrated in northern China, which has been a hot spot for wind erosion research [31–33]. However, since the scarcity of associated spatial data, especially the spatial distribution data of ecological programs, these studies, referring to the wind erosion driving factors in northern China, have mostly concentrated on the qualitative assessment of the effects of wind speed, temperature, and other single indicators on the annual wind erosion modulus [15,30]. Moreover, they have not considered the effects of the multiple determinants (such as ecological programs) that have altered the spatiotemporal variation of soil wind erosion on the regional scale [31–33].

The objectives of this study were to (1) estimate soil wind erosion in the TNR from 1981 to 2020 based on RWEQ and depict its spatial pattern and temporal dynamics; (2) explore the effects of climatic factors, such as temperature, precipitation, and wind speed, on soil wind erosion at pixel scale; and (3) determine the effects of ecological programs and human interference on soil wind erosion, then comprehensively analysis their effects in wind erosion at the pixel scale.

## 2. Materials and Methods

### 2.1. Study Area

The total area of China's TNR is 4.07 million km², located between 73°26′E and 127°50′E and 33°30′N and 50°12′N. The TNR comprises 551 counties in 13 provinces, occupying over 42.4% of China's entire land territory (Figure 1). It is divided into four zones—Northeast, North Central, Loess Plateau, and Northwest—according to geomorphological and climatic characteristics.

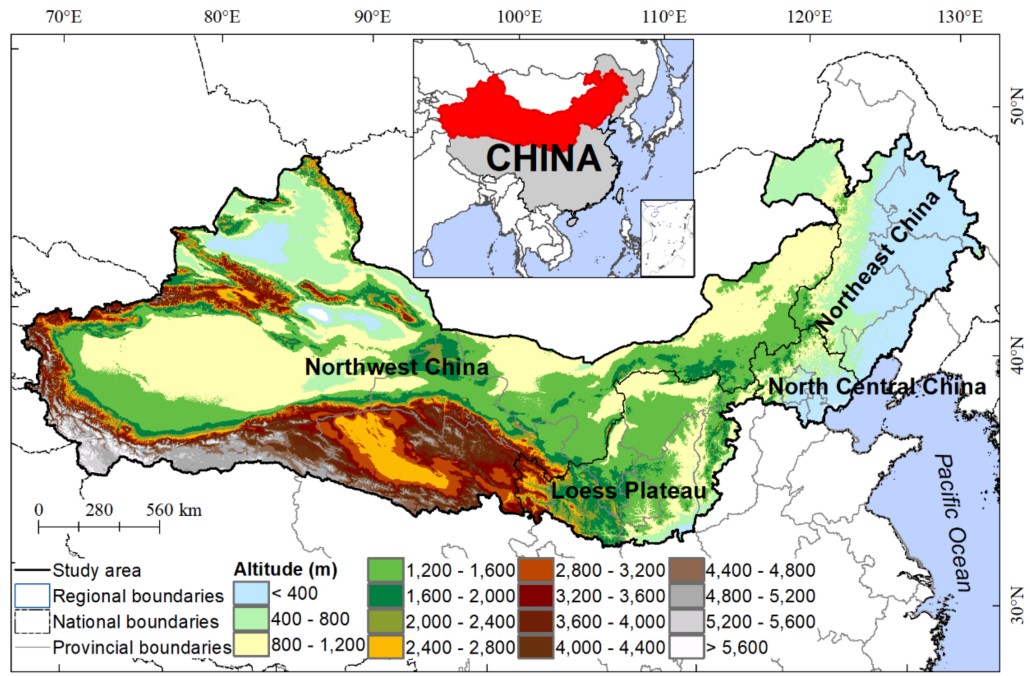

**Figure 1.** Location of China's Three-North Region (TNR).

Northeast and North Central China are affected by a temperate monsoon climate, while the Loess Plateau and Northwest China are mainly affected by a temperate continental climate. Temperatures vary widely throughout the TNR, with annual mean temperatures ranging from approximately 15 °C in the south to approximately −17 °C in the mountains of the northwest, with temperatures ranging from 3 to 9 °C in most areas. The precipitation diminishes from the east (approximately 750 mm) to the west (less than 50 mm), which reflects the decreasing effect of moist air from the Pacific Ocean. Strong winds occur very often in the winter and spring, with an annual mean wind velocity of approximately 1.7 m/s to 2.3 m/s and a maximum wind velocity of up to 20 m/s. Sand dust and sandstorms usually occur 20–30 days each year within the region. The TNR of China has three major geomorphic units: high mountains (e.g., Mount Kunlun and Mount Tianshan), basins (e.g., the Tarim and Qaidam Basins), and plains (e.g., the North China and Northeast China Plains), and the elevations range from approximately 155 m under sea level at Ayding Lake to greater than 7000 m high at Mount Kunlun (see Supplementary Figure S1) [10–12]. The TNR has a variety of soil types, and the soil types are, in order from east to west, as follows: black soil, brown soil, gray–brown desert chestnut soil, and aeolian sandy soil. The study area's vegetation ranges from forests in the east (e.g., broad-leaved forests and needle forests) to barren land and sandy land (e.g., shrubs and sparse grassland), desert, and grassland toward the west.

### 2.2. Data Collection and Processing

Daily meteorological data, collected from the China Meteorological Data Service, include wind speed, solar radiation, precipitation, and temperature. In addition, 255 ground metrological stations were available in the study. The weather factors were

spatially interpolated based on the ANUSPLIN (2018, https://fennerschool.anu.edu.au/research/products/anusplin, accessed on 10 August 2022) model by combining meteorological data and a digital elevation model (DEM). DEM of 30 m spatial resolution was obtained from the Geospatial Data Cloud (2018, http://www.gscloud.cn/, accessed on 10 August 2022).

Soil data (soil type and soil fractions) were obtained from the China Soil Database (2018, https://data.tpdc.ac.cn/zh-hans/, accessed on 10 August 2022) of the National Tibetan Plateau Data Center (spatial resolution: 1 km). The soil database in China was come from the Institute of Nanjing Soil science and used the FAO-90 to categorize the soil types. Here, we use the top 30 cm soil database, including sand content, silt content, clay content and $CaCO_3$ that will be used in the RWEQ.

Normalized difference vegetation index (NDVI) data, including AVHRR (Advanced Very High-Resolution Radiometer) and GIMMS (Global Inventory Modeling and Mapping Studies) NDVI 3 g products, were available from 1981 to 2000, and Terra MODIS (Moderate Resolution Imaging Spectroradiometer) NDVI (MOD13A3) data were available from 2001 to 2020. Due to the difference between the NOAA NDVI and MODIS NDVI sensors, a statistical regression model between AVHRR NDVI and MODIS NDVI were established pixel by pixel using two data series during the overlapped period (2001–2006). Then, AVHRR NDVI back to 1981 was corrected from historical AVHRR observations based on these pixel-level relationships. The long-term NDVI series was created by a combination of AVHRR NDVI (1981–2000) and MODIS NDVI (2001–2020).

All data were harmonized and transformed into 1 km × 1 km grid data.

### 2.3. Revised Soil Wind Erosion Model

Following previous studies [4,34–37], the RWEQ was employed to model the wind erosion moduli based on the weather, soil, surface roughness, and vegetation factors. The primary equations of the RWEQ are as follows:

$$SL = \frac{2z}{S^2} \times Q_{max} \times e^{-\left(\frac{z}{s}\right)^2} \tag{1}$$

$$TC_{max} = 109.8\left(WF \times SEF \times SCF \times K' \times VF\right) \tag{2}$$

$$s = 150.71 \times \left(WF \times SEF \times SCF \times K' \times VF\right)^{-0.3711} \tag{3}$$

where *SL* is the soil wind erosion flux (kg m$^{-2}$ a$^{-1}$); *z* is the distance from upwind distance from the maximum wind erosion (m), usually 50 m; $TC_{max}$ is the maximum transport capacity of the soil wind erosion (kg m$^{-1}$); and s is where 63% of the maximum transmission capacity occurs, which is generally called the critical field length. *WF* is the weather fraction (kg m$^{-1}$), *SEF* is the soil erodibility fraction (nondimensional), *SCF* is the surface crust fraction (nondimensional), $K'$ is the soil roughness fraction (nondimensional), and *VF* is the vegetation fraction (nondimensional). Fryrear et al. [21], Du et al. [33] and Jiang [35] provided detailed formulas for each fraction.

For the verification of wind erosion results from the RWEQ, we collected two types of ground monitoring data: one part was ground monitoring data based on sand trap tools from 2018 to 2020, and the other part was from references at the field scale based on META analysis (Table 1). To obtain the amount of wind erosion modulus in Wengniuot Banner (a county located in western Horqin Sandy Land), we added all weights of trapped sand and then divided by 4. Based on 25 groups of data, we found a good correlation between the measured data and the simulation results ($R^2$ = 0.77, $p < 0.01$) (Figure 2a).

**Table 1.** Wind erosion modulus from published papers and field observations.

| ID | Name | Location | Year | Wind Erosion Modulus from References and Ground Monitoring (t km$^{-2}$) | Simulated Wind Erosion Modulus (t km$^{-2}$) |
|---|---|---|---|---|---|
| 1 | Zhengxiangbai Qi | 115.55E, 42.33N | 2007 | 351.00 | 339.54 |
| 2 | Xilinghot | 116.10E, 43.76N | 2007 | 360.00 | 167.40 |
| 3 | Taipusi Qi 1 | 115.49E, 42.11N | 2008 | 418.00 | 70.06 |
| 4 | Taipusi Qi 2 | 115.17E, 41.76N | 2008 | 480.00 | 56.90 |
| 5 | Taipusi Qi 3 | 115.34E, 41.96N | 2008 | 310.00 | 66.23 |
| 6 | Geermi s1 | 94.92E, 36.42N | 1997 | 8414.00 | 4965.99 |
| 7 | zhd 1 | 88.60E, 44.39N | 2015 | 739.66 | 1341.49 |
| 8 | zhd 2 | 88.60E, 44.37N | 2015 | 945.06 | 1308.22 |
| 9 | zhd 3 | 89.20E, 45.05N | 2015 | 4404.01 | 1433.64 |
| 10 | zhd 4 | 89.17E, 44.66N | 2015 | 3538.01 | 2260.93 |
| 11 | zhd 5 | 89.61E, 44.79N | 2015 | 1720.08 | 2167.11 |
| 12 | zhd 6 | 89.61E, 44.27N | 2015 | 3644.22 | 1447.71 |
| 13 | zhd 8 | 90.35E, 44.27N | 2015 | 2644.08 | 1706.66 |
| 14 | zhd 9 | 88.59E, 45.11N | 2015 | 3855.57 | 1627.36 |
| 15 | zhd10 | 89.11E, 44.24N | 2015 | 749.35 | 1453.61 |
| 16 | zhd 11 | 89.91E, 44.50N | 2015 | 1871.13 | 1616.11 |
| 17 | zhd 12 | 89.35E, 44.79N | 2015 | 1840.09 | 1776.69 |
| 18 | zhd 13 | 90.09E, 44.32N | 2015 | 1437.38 | 1525.16 |
| 19 | Bayannaoer 1 | 104.47E, 41.80N | 2006 | 64.58 | 150.85 |
| 20 | Halahelin | 103E, 41.10N | 2006 | 6723.06 | 5500.07 |
| 21 | Wuchuan County | 111.21E, 41.21N | 2018 | 2160.17 | 102.78 |
| 22 | Bashang Region | 114.35E, 41.90N | 2005 | 8362.34 | 6780.01 |
| 23 | Gonghe Basin | 100E, 36.10N | 1999 | 2491.00 | 341.15 |
| 24 | Zhd | 90E, 44N | 2016 | 4345.99 | 1383.63 |

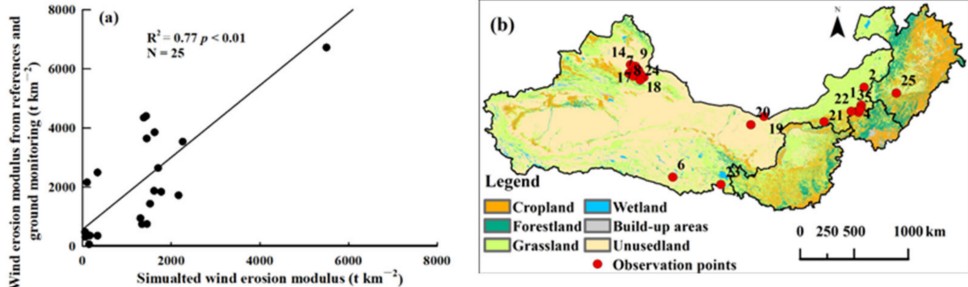

**Figure 2.** (**a**) Model accuracy test of the wind erosion modulus and (**b**) distribution of field points across the TNR of China (the number of the observation point is the ID of Table 1).

Following the Soil Erosion Classification and Grading Standard (SL 190–2007) issued by the Chinese government, soil wind erosion in the TNR from 1981 to 2020 was classified into six intensity levels: tolerable, mild, moderate, severe, more severe, and very severe (Table 2).

**Table 2.** Classification of the intensity of soil wind erosion in the TNR by the PRC Ministry of Water Resources (SL190−2007).

| Levels | Vegetation Coverage (%) | Wind Erosion Modulus (t/(km$^2$/a)) |
|---|---|---|
| Tolerable erosion | >70 | <200 |
| Mild erosion | 50–70 | 200–2500 |
| Moderate erosion | 30–50 | 2500–5000 |
| Severe erosion | 10–30 | 5000–8000 |
| More severe erosion | <10 | 8000–15,000 |
| Very severe erosion | <10 | >15,000 |

*2.4. Statistical Analysis*

2.4.1. Trend Analysis

For the trend statistics, we used the ordinary least squares (OLS) approach to monitor the long-term sequential dynamic change in soil wind erosion and its influencing factors (i.e., climate and ecological programs) in different periods. This method comprehensively reflects the temporal and spatial patterns of a region by calculating the temporal change at the pixel scale. The formula of the slope is as follows:

$$SL_{slope} = \frac{n \times \sum_{t=1}^{n} t \times SL_i - \sum_{t=1}^{n} t \sum_{t=1}^{n} SL_t}{n \times \sum_{t=1}^{n} t^2 - \left(\sum_{t=1}^{n} t\right)^2} \tag{4}$$

where $SL_{slope}$ represents the gradient of the trend; $n$ represents the whole years from 1981 to 2020; $t$ is the serial number from 1 to $n$; and $SL_i$ is the amount of wind erosion in the $i$th year. The trend of soil wind erosion and its influencing factors was classified into five classes:

Significant increase (SI): a significant positive trend ($SL_{slope} \geq 0.001$, $p \leq 0.05$);
Nonsignificant increase (NI): a nonsignificant positive trend ($SL_{slope} \geq 0.001$, $p > 0.05$);
Nonsignificant decrease (ND): a nonsignificant negative trend ($SL_{slope} \leq -0.001$, $p > 0.05$);
Significant decrease (SD): a significant negative trend ($SL_{slope} \geq 0.001$, $p \leq 0.05$);
No change (NC): no trend with a significant tendency ($-0.001 < SL_{slope} < 0.001$).

2.4.2. Driving Force Analyses

Wind erosion is affected by multiple factors, and the influencing factors interact with each other. Based on previous studies, we selected three types of driving factors: ecological programs, climate factors, and human interference (Table 3).

**Table 3.** Driving factors and data sources.

| First-Level Indicators | Second-Level Indicators | Data Source and Processing | Data Accuracy |
|---|---|---|---|
| Ecological programs | Area porportion of forest and shrubland in a 1 km × 1 km grid | Visual interpretation using multiphase remote sensing images (Zheng and Zhu 2017) | 0.95 |
| | LAI of forest and shrubland | MODIS LAI (2001–2020) and LAI (1981–1999) were calculated based on the relationship between NOAA-NDVI and MODIS-LAI | 0.71 |
| | NPP of forest and shrubland | CASA model | 0.82 |
| Human interference | Area proportion of cropland in a 1 km × 1 km grid | Same as the area of forest and shrubland | 0.95 |
| | Area proportion of grassland in a 1 km × 1 km grid | Same as the area of forest and shrubland | 0.95 |
| Climate changes | Spring mean wind speed | Spatialization of meteorological data using the ANUSPLIN and cokriging interpolation method | 0.8 |
| | Spring mean temperature | | 0.83 |
| | Spring precipitation | | 0.85 |

The ecological programs are a series of ecological restoration programs. In addition, we found that the forest and shrubland area was increasing mainly due to the ecological programs' complement, based on the field survey and national forest inventory data. So, here, the changes in forest and shrubland in the area, leaf area index (LAI), and net primary productivity (NPP) were used as indicators of the ecological programs. The spatiotemporal dynamics of forest and shrubland were derived by visual interpretation based on Landsat images from 1981 to 2020. LAI data from 2001 to 2020 were generated from MODIS data (MODIS15A2), and those from 1981 to 2000 were estimated from historical AVHRR observations based on the relationship between AVHRR GIMMS NDVI and MODIS LAI during the overlapping period (2000–2010). Annual NPP was used and simulated using the Carnegie-Ames-Stanford Approach (CASA) model with NDVI sequences derived from AVHRR GIMMS (1984–2000) and Terra MODIS products (2000–2020) [38]. For validation of the NPP results, the coefficient of correlation between the annual NPP derived from MOD17A3 and CASA-modeled NPP was 0.82.

For human interference, the changes in the area proportion of cropland grassland and in a 1 km × 1 km grid were selected. They were also derived by visual interpretation first, and then the method was the same as that used for the area proportion of forest and shrubland in a 1 km × 1 km grid.

Climate changes (e.g., wind, precipitation, and temperature) play a critical role in soil moisture, hydrological processes, and vegetation growth [1,39,40], so the role of wind erosion changes is also extremely significant. In northern China, wind erosion during late winter and spring accounts for more than 80% of the annual wind erosion; thus, we chose the climate changes from February to May (recorded as winter–spring in this study) to analyze the effects on the variation in wind erosion [8,41]. Average wind speed plays an essential role in altering wind erosion dynamics, and the interactive effects of strong winds combined with the dried, loosened soil surface, and sparse vegetation coverage could induce serious wind erosion [40,42]. Precipitation in the winter–spring season could alter the vegetation condition and soil moisture. The temperature in the winter–spring season can reduce snow cover and soil moisture. Together, the above two aspects play a role in wind erosion. Therefore, we used wind speed, precipitation, and mean temperature in the winter–spring season to reveal the effects of the various climatic factors on wind erosion in the TNR of China. According to preliminary work, the spring precipitation was spatially interpolated by using the cokriging method; the others were interpolated by using the ANUSPLINE model.

The Pearson correlation method [1,43] was applied to analyze the correlation between the dynamics of wind erosion and the changes in ecological programs, climate factors, and human interference, while Student's t-test was used as the evaluation criterion for the significance of the correlation coefficients. The correlation between soil wind erosion and its driving factors was classified into five classes:

Significant positive correlation (SP): $COcoef \geq 0.2$, $p \leq 0.05$;
Nonsignificant positive correlation (NP): $COcoef \geq 0.2$, $p > 0.05$;
Nonsignificant negative correlation (NN): $COcoef \leq -0.2$, $p > 0.05$;
Significant negative correlation (SN): $COcoef \leq -0.2$, $p \leq 0.05$;
Uncorrelated (UC): $-0.2 < COcoef < 0.2$.

To determine the contribution of ecological programs and other factors (e.g., climate factors and human interference) to wind erosion (i.e., comprehensive analysis or multifactor analysis), we used multiple linear regressions. Stepwise multiple linear regression is used to construct some purposely chosen explanatory variables for the dependent variable $Y$. The equations are picked from the maximum multiple correlation coefficient. The statistical significance of the contributing variables was $p < 0.05$. To facilitate the comparison of contributions of different variables in the stepwise multiple linear regression models, all variables with units are standardized [44]. In the multiple linear standardized regression model, the importance of the independent variables is calculated with the following equation:

$$Y = a_1 X_1 + a_2 X_2 + \cdots + a_n X_n \tag{5}$$

$$\eta = |a_\mathrm{m}| / \sum a_\mathrm{m} \tag{6}$$

where $Y$ is the dependency variable (that is, the wind erosion modulus), $X_i$ is the independent variable (Table 3), and $a_\mathrm{m}$ ($1 \leq \mathrm{m} \leq n$) is the standard partial regression coefficient for variable m in the stepwise multiple linear regression equation, indicating the amount of change in the responsive variable $Y$ when the explanatory variable varies by one unit. $\eta$ is the relative importance. The data extraction process is as follows. First, the study area was divided into 1 km × 1 km sample areas, as statistical units. Second, we extracted the changes in wind erosion amount in each sample area, and the corresponding changes in ecological programs, climate changes, and human interference. Third, the stepwise multiple linear regression was used to illustrate the driving force of wind erosion at the pixel scale (i.e., 1 km × 1 km).

## 3. Results

### 3.1. Spatial Pattern of Wind Erosion in the TNR

The soil wind erosion in the TNR in China from 1981 to 2020 was obtained based on RWEQ. In the TNR, the mean annual wind erosion was approximately 25 billion tons from 1981 to 2020. Following the Soil Erosion Classification and Grading Standard (SL190−2007) released by the Ministry of Water Resources of the People's Republic of China (Table 2), tolerable erosion, slight erosion, and moderate erosion were the main kinds of wind erosion within the TNR. The area of tolerable erosion was approximately 1.42 million km$^2$ (37.67% of the overall area of TNR), primarily distributed in areas with high vegetation coverage and low soil sandy content, such as the Northeast China Plain (see Supplementary Figure S1), North Central China, and the southern Loess Plateau. The area of slight erosion encompassed 1.16 million km$^2$ or 30.94% of the overall area of the TNR, and it mostly occurred at the edge of the sandy land with relatively high vegetation coverage and relatively high sandy soil content. Moderate erosion was mainly distributed in four sandy lands (e.g., Hulunbuir Sandy Land, Horqin Sandy Land, Mu Us Sandy Land, and Otindag Sandy Land) (see Supplementary Figure S1), and the content of sand in the soil was relatively high. The severe erosion area was approximately 0.23 million km$^2$, or 3.60% of the overall area of the TNR, and it was primarily located in the Gobi area. The total area of more severe and extremely severe erosion was approximately 0.82 million km$^2$ and was mainly located in the large deserts (e.g., the Taklimakan Desert, the Gurbantunggut Desert, the Tengger Desert, the Badain Jaran Desert, the Kumtag Desert, the Qaidam Basin Desert, the Kubuqi Desert, and the Ulanbuhe Desert) (see Supplementary Figure S1), which are the main sources of sandstorms in China (Figure 3).

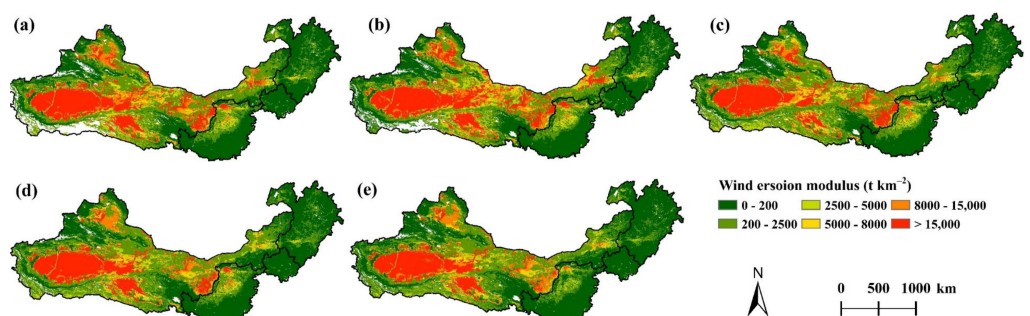

**Figure 3.** Spatial pattern of the mean wind erosion modulus in the TNR of China from 1981 to 2020. (**a**) 1981–2020; (**b**) 1981–1990; (**c**) 1991–2000; (**d**) 2001–2010; (**e**) 2011–2020.

### 3.2. Spatiotemporal Variations in Wind Erosion

In the 40-year simulation in the TNR, overall, wind erosion showed a decreasing trend from 1981 to 2020, with the average wind erosion modulus decreasing from 10,016.65 t km$^{-2}$ in 1981 to 5268.84 t km$^{-2}$ in 2020, representing an annual average decrease of 99.02 t km$^{-2}$. The change in wind erosion in the TNR could be divided into four phases. There was a significant decreasing trend in the wind erosion modulus (slope = −302.04 t km$^{-2}$ a$^{-1}$, $p < 0.01$) from 1981 to 1990, an insignificant increase from 1990 to 2001, a significant decrease with a slope of 200.53 t km$^{-2}$ a$^{-1}$ from 2001 to 2014 ($p < 0.01$), and then an insignificant increase from 2014 to 2020 (Figure 4a).

According to the Student's t-test, 40.21% of the TNR had a decreasing trend, chiefly in the central and northern TNR of China. Within the region, approximately 19.08% showed a significant decrease in the mean wind erosion modulus, which was mainly distributed in the deserts and sandy lands, except for the Taklimakan Desert ($p < 0.05$) (see Supplementary Figure S1). A total of 15.77% of the TNR of China showed an increasing trend during the last 40 years, which was mainly distributed in the western TNR of China. Approximately 75.00% of the increasing region of the mean wind erosion modulus was significant, which was mainly distributed in the Tarim Basin. During the past 40 years, approximately 44.01%

of the TNR of China showed a nonsignificant change trend in the wind erosion modulus, which was widely distributed in the region (Figure 4b).

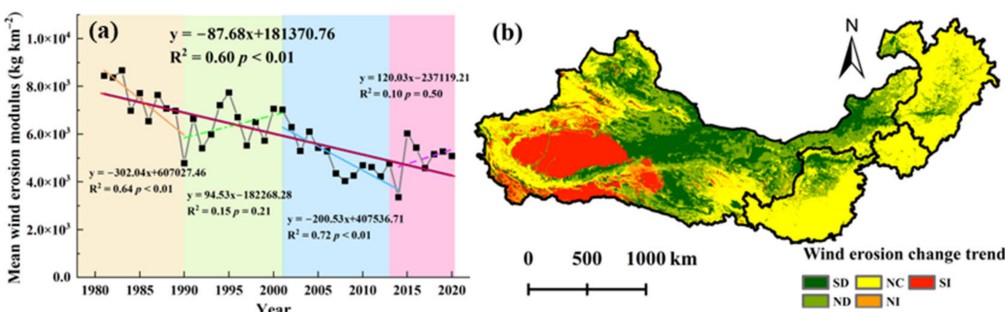

**Figure 4.** Variation in the wind erosion modulus in the TNR of China from 1981 to 2020. (**a**) Temporal change and (**b**) spatial pattern variation in the wind erosion modulus in the TNR of China from 1981 to 2020. SD: significant decrease; ND: nonsignificant decrease; NC: no change; NI: nonsignificant increase; SI: significant increase.

According to geomorphological and climatic characteristics, the TNR of China is divided into four subregions: Northeast China, North Central China, the Loess Plateau, and Northwest China (Figure 1). We analyzed the mean wind erosion modulus dynamics in the four subregions. Northeast China mainly consisted of the Northeast China Plain, Horqin Sandy Land, and Hulunbuir Sandy Land (see Supplementary Figure S1), and the mean wind erosion modulus had a significant decreasing trend, from 892.27 t km$^{-2}$ a$^{-1}$ in 1981 to 432.52 t km$^{-2}$ in 2020, with an average decline rate of 9.03 t km$^{-2}$ a$^{-1}$ (Figure 5a). According to the Student's t-test, 16.33% (approximately 95,373 km$^2$) of Northeast China showed a decreasing trend, 82.79% of the region showed a nonsignificant change, and only a few parts (0.16%) exhibited a significant growth trend over the last 40 years (Figure 4b).

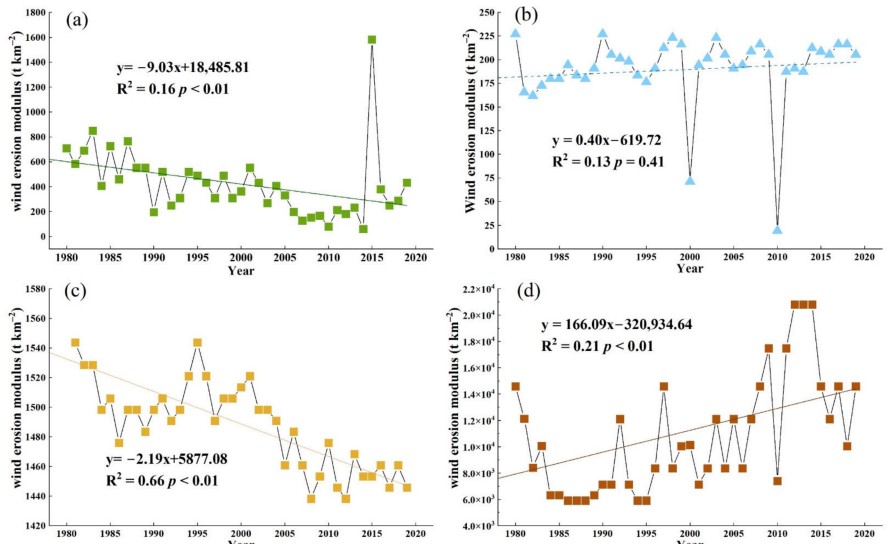

**Figure 5.** Dynamic wind erosion modulus in (**a**) Northeast China, (**b**) North Central China, (**c**) the Loess Plateau, (**d**) and Northwest China from 1981 to 2020.

North Central China is in the northern North China Plain (see Supplementary Figure S1), and the annual mean wind erosion modulus showed an insignificant increasing trend (Figure 5b). Most of the region (94.26%) showed a nonsignificant change during the past 40 years (Figure 4b); 0.57% of the region showed an increasing trend, and approximately 5.17% of the region showed a significant decreasing trend on a small scale, although most of these sites were distributed in northern North Central China.

The Loess Plateau is the largest loess deposition area in the world and is situated in the central TNR. In this region, the average wind erosion modulus showed a significant decreasing trend, from 1528.56 t km$^{-2}$ a$^{-1}$ in 1981 to 1445.72 t km$^{-2}$ a$^{-1}$ in 2020, representing an annual mean rate of 2.19 t km$^{-2}$ (Figure 5c). Based on the trend analysis, 21.55% of the area showed a clear downward trend, mostly in the northern part of the Loess Plateau (Figure 4b). A total of 1.10% of the region had an upward trend, mostly in the western Loess Plateau. However, most of the region (77.36%) showed nonsignificant changes during the past 40 years.

Northwest China has the largest area of the TNR, and all of China's deserts (see Supplementary Figure S1) are in this region. In this region, the average wind erosion modulus increased significantly from 12,138.77 t km$^{-2}$ in 1981 to 14,518.73 t km$^{-2}$ in 2020, representing a mean rate of 166.09 t km$^{-2}$ a$^{-1}$ (Figure 5d). In addition, 22.36% of the region had an upward trend, mostly in western Northwest China (Figure 4b). In contrast, 50.39% of Northwest China showed a decreasing trend during the past 40 years. Only 27.24% of Northwest China showed a nonsignificant change trend.

### 3.3. Spatiotemporal Variations in Driving Force Factors

#### 3.3.1. Changes in Forest and Shrubland Areas Influenced by Ecological Programs

Since the implementation of ecological programs, the total forest and shrubland area, derived from Landsat satellite remote sensing images, has increased by 164,605 km$^2$ with a slope of 41,151.25 km$^2$ decade$^{-1}$ (from 214,001 km$^2$ in 1981 to 378,606 km$^2$ in 2020); this is a change of 76.92%, from 5.26% of the total land cover in 1981 to 9.30% in 2020 (Figure 6a). In addition to forest area, forest quality is also a key factor in analyzing the impact of ecological programs on wind erosion (Table 4). We selected the LAI and NPP as the proxies of forest quality. The LAI of the ecological programs showed a significant increasing trend, and approximately 135,408 km$^2$ had a significant increase and was distributed in northern Northeast China, North Central China, and the Loess Plateau (Figure 6b). The nonsignificant change area was 19,107 km$^2$, which was mainly distributed in the western Loess Plateau. The decrease in the LAI was 134 km$^2$ and occupied only a very small part of the ecological programs. The average NPP value increased from 68.08 g C km$^{-2}$ in 1981 to 69.09 g C km$^{-2}$ in 2020. Based on the tendency of NPP in forest and shrubland, an area of approximately 110,318 km$^2$ showed a significant increasing trend, which was mostly in Northeast China, North Central China, the western Loess Plateau, and northern Northwest China; in contrast, the areas with nonsignificant change and a significant decreasing trend were 62,774 km$^2$ and 152,828 km$^2$, respectively (Figure 6c).

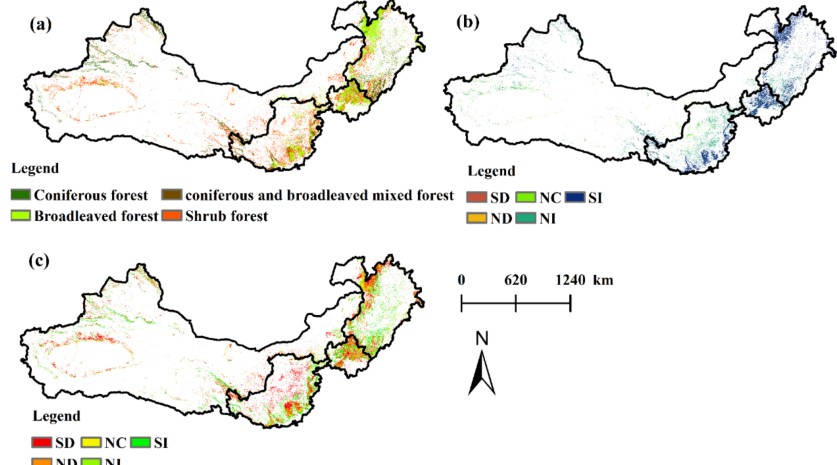

**Figure 6.** Distribution of forest type and dynamic change in forest quality in ecological programs. (**a**) Distribution of forest type; dynamic change in (**b**) leaf area index (LAI) of the forest and (**c**) net primary production (NPP) of the forest. SD: significant decrease; ND: nonsignificant decrease; NC: no change; NI: nonsignificant increase; SI: significant increase.

**Table 4.** Changes in the forest and shrubland area during 1981 and 2020 (km$^2$).

| Year | Coniferous Forest | Broad-Leaved Forest | Coniferous and Broad-Leaved Forest | Shrubland | Total |
|------|------|------|------|------|------|
| 1981 | 38,098 | 87,861 | 17,454 | 70,587 | 214,001 |
| 1990 | 57,988 | 105,992 | 23,505 | 103,491 | 290,976 |
| 2000 | 59,493 | 114,577 | 21,591 | 114,577 | 310,238 |
| 2010 | 49,345 | 110,095 | 20,132 | 165,192 | 344,763 |
| 2020 | 45,536 | 118,817 | 15,135 | 199,118 | 378,606 |

### 3.3.2. Climate Changes

The average temperature in late winter–spring in the TNR of China increased from 8.73 °C in 1981 to 10.36 °C in 2020 and increased significantly at a rate of 0.503 °C a$^{-1}$ ($R^2$ = 0.63, $p$ < 0.01) (Figure 7a). Spatially, the warming in eastern Northeast China and Mount Qilian in Northwest China was relatively limited, while the warming in North Central China and most of Northwest China was relatively widespread (Figure 7b). Precipitation in late winter–spring showed an increasing trend in most areas but was not significant (Figure 7c, d). From 1981 to 2020, the average wind speed in late winter–spring had a distinct decreasing trend of 0.0145 m s$^{-1}$ a$^{-1}$ ($R^2$ = 0.63, $p$ < 0.01) (Figure 7e). The decrease in wind speed increased from east to west, with the higher elevations in western Northwest China showing an increase in wind speed (Figure 7f).

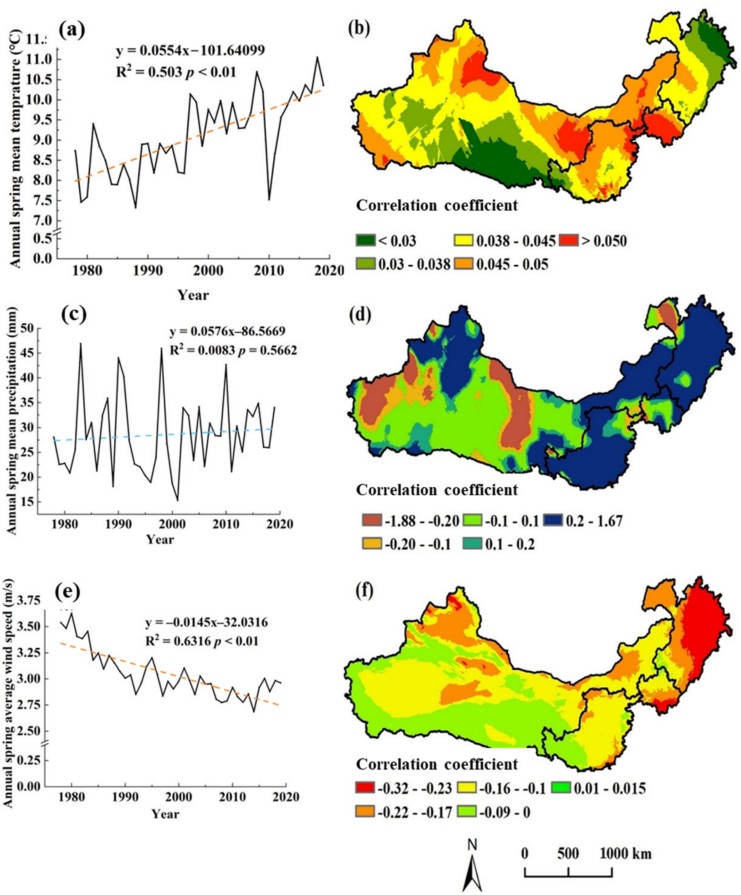

**Figure 7.** Dynamic change in the climate changes in the TNR of China from 1981 to 2020. (**a**) Temporal change and (**b**) spatial pattern change in mean temperature; (**c**) temporal change and (**d**) spatial pattern change in precipitation; (**e**) temporal change and (**f**) spatial pattern change in mean wind speed.

### 3.3.3. Changes in Human Interference

As a result of socioeconomic development, the area proportion of grassland and cropland in a 1 km × 1 km grid has changed significantly during the past 40 years.

The area proportion of grassland in a 1 km × 1 km grid decreased by 65,966 km$^2$ with a slope of 6596.60 km$^2$ decade$^{-1}$ from 1,388,888 km$^2$ in 1981 to 1,322,922 km$^2$ in 2020. The unchanged area proportion of grassland in a 1 km × 1 km grid was mainly distributed in Northeast China, North Central China, and the Loess Plateau (Figure 8a). The decreased area proportion of grassland in a 1 km × 1 km grid was mainly located on the Loess Plateau.

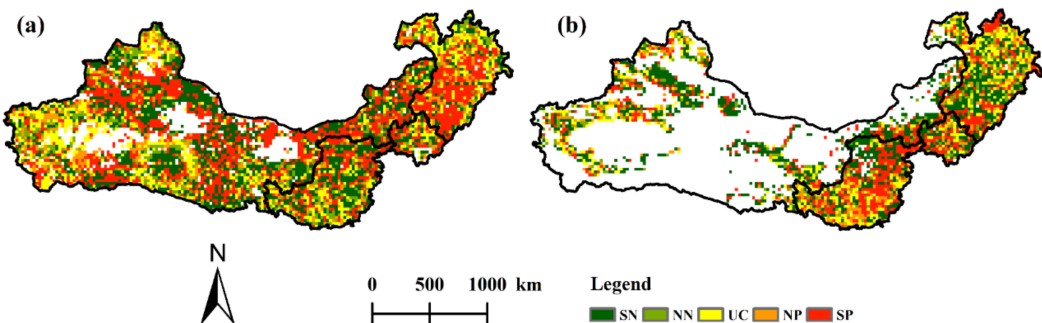

**Figure 8.** Spatiotemporal changes in (**a**) area proportion of grassland in 1 km × 1 km grid and (**b**) area proportion of cropland in 1 km × 1 km grid from 1981 to 2020.

The area proportion of cropland in a 1 km × 1 km grid increased by 45,575 km$^2$ with a slope of 1139.38 km$^2$ decade$^{-1}$ (from 514,681 km$^2$ in 1981 to 560,256 km$^2$ in 2020). The increased area proportion of cropland in a 1 km × 1 km grid was mainly distributed in western Northeast China and western Northwest China (Figure 8b). The unchanged area proportion of cropland in a 1 km × 1 km grid was mainly distributed in central Northeast China, southern North Central China, and the southern Loess Plateau. The decrease in area proportion of cropland in a 1 km × 1 km grid mainly occurred in the unchanged area.

## 4. Discussion

### 4.1. Correlations between Ecological Programs and Wind Erosion

In general, afforestation will increase the coarseness of the land surface, alter the water content of the soil, and, thus, promote a reduction in wind erosion, which is one of the purposes of the construction of ecological programs [12,45,46]. From 1981 to 2020, the area proportion of forest and shrubland contributed significantly to the reduction in soil wind erosion, affecting 25% of the TNR (Figure 9a). A negative correlation was found to exist between soil wind erosion and the area proportion of forest and shrubland across 37.45% of the study area, particularly in North Central China, the northern Loess Plateau, and eastern Northwest China, where more than 22% of the area proportion of forest and shrubland was unlikely to experience wind erosion (Figure 9a). In addition, the quality of forest and shrubland (the combined index calculated by LAI × NPP) was negatively and significantly correlated with soil wind erosion and affected approximately 22.07% of the study area (Figure 9b). The increased area proportion and quality of forests and shrubs reduced the areas of significant wind erosion in Northeast China, North Central China, the northern Loess Plateau, and eastern Northwest China [47]. No apparent correlation was found between soil wind erosion and the area proportion and quality of forest and shrubland in 67.62% and 64.10% of the TNR of China, respectively. In summary, the ecological programs experienced a significant reduction effect of wind erosion, but the effect was relatively limited.

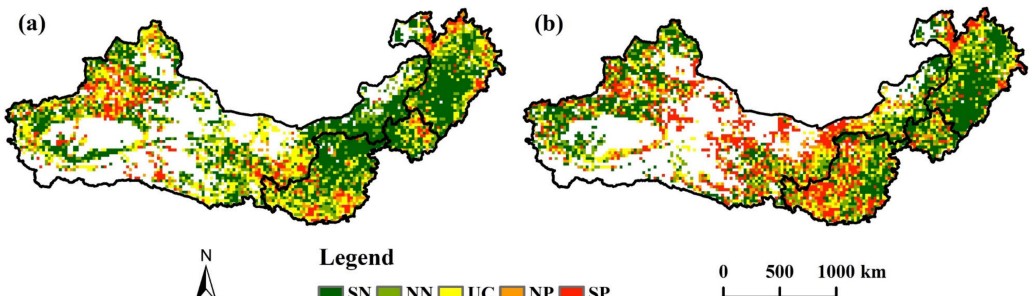

**Figure 9.** Spatial correlations between wind erosion and (**a**) area proportion of the forest and shrubland and (**b**) quality of the forest and shrubland (the quality index calculated by LAI × NPP). SN: significant negative correlation; NN: nonsignificant negative correlation; UC: uncorrelated; NP: nonsignificant positive correlation; SP: significant positive correlation.

### 4.2. Correlations between Climatic Factors and Wind Erosion

According to the relationship between the changes in climatic variables, including spring mean wind speed, spring precipitation, and spring mean temperature, and the changes in wind erosion, an obvious positive correlation ($p < 0.05$) existed between the spring mean wind speed and wind erosion across 62.83% of the TNR (Figure 10c), which covered almost all Northeast China and Northwest China. Wind speed can change the spatial patterns and temporal variations in wind erosion. For example, a slight increase in the average spring wind speed in 1993–1994 and 1998–2000 led to an increase in the annual mean wind erosion modulus (Figures 4a and 7c) [48]. Therefore, in the TNR, the wind carried soil, transported sand, and directly affected wind erosion; thus, the spring wind speed was one of the most important factors that dominated the change in wind erosion [48].

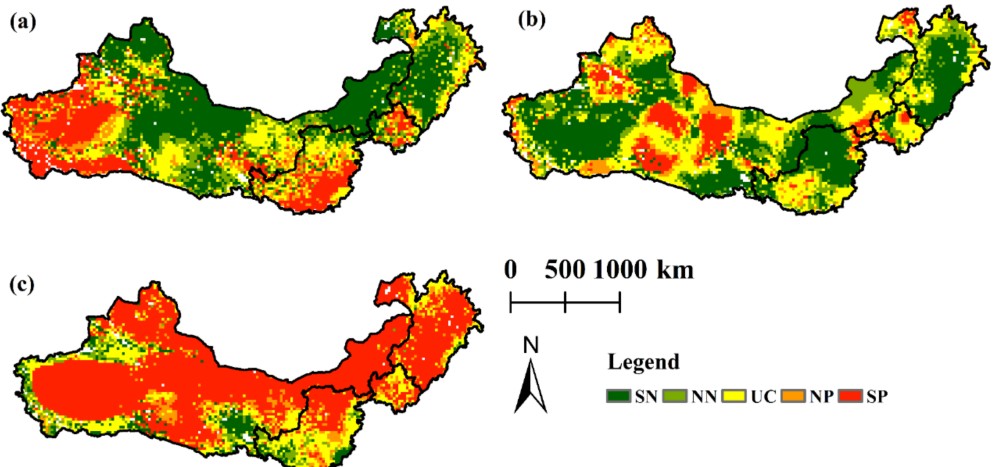

**Figure 10.** Spatial correlations between wind erosion and (**a**) spring mean temperature, (**b**) spring precipitation, and (**c**) spring average wind speed (SN: significant negative correlation; NN: nonsignificant negative correlation; UC: uncorrelated; NP: nonsignificant positive correlation; SP: significant positive correlation).

In terms of temperature, approximately 80.42% of the TNR was affected by the spring average temperature from 1981 to 2020, and 20% of the TNR presented a distinct positive correlation ($p < 0.05$) (Figure 10a). Therefore, the high temperature played a key role in limiting the growth of vegetation, thus increasing soil wind erosion to some extent in western Northwest China.

The area with a negatively correlated relationship between soil wind erosion and precipitation occupied 44.48% of the total area (Figure 10b), particularly in eastern Northeast China, the northern Loess Plateau, and southwestern Northwest China. Arid and semiarid

areas receive less than 30 mm of spring precipitation and are, therefore, more susceptible to wind erosion. In addition, increased precipitation can indirectly affect wind erosion by altering the soil water content, land surface coverage, and vegetation growth [36,42,49]. A total of 19.58% of soil wind erosion was not significantly correlated with temperature, and 18.48% of soil wind erosion and precipitation were not significantly correlated.

### 4.3. Correlations between Human Interference and Wind Erosion Change

In the TNR of China, the effect of human interference on wind erosion is much smaller than that of climatic variables [36]. Here, we selected two factors that affected wind erosion in the TNR of China, including the area proportion of cropland and grassland in a 1 km × 1 km grid [50,51].

The change in the area proportion of grassland contributed to wind erosion between 1981 and 2020, and wind erosion significantly negatively affected approximately 30.28% of the TNR of China (Figure 11a). Wind erosion increased as the area proportion of grassland decreased. The wind erosion area affected by the area proportion of grassland was mostly located in western Northeast China and northern Northwest China.

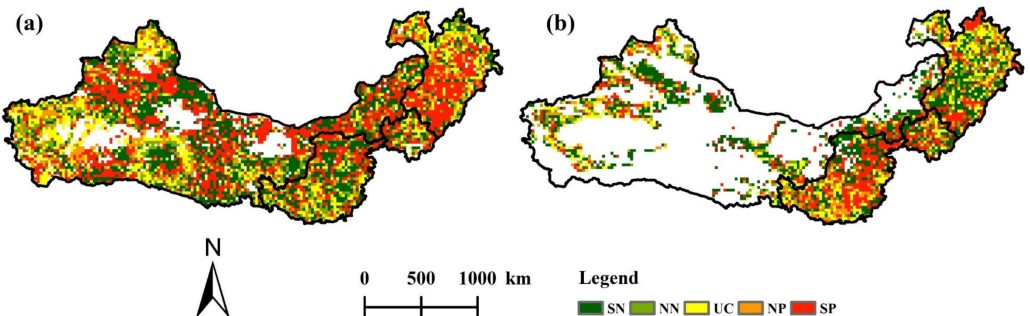

**Figure 11.** Spatial correlations between wind erosion and (**a**) intensity of grazing activities and (**b**) agricultural activities (SN: significant negative correlation; NN: nonsignificant negative correlation; UC: uncorrelated; NP: nonsignificant positive correlation; SP: significant positive correlation).

Over 13.41% ($p < 0.05$) of the study area, a negative correlation between soil wind erosion and the area proportion of cropland was observed (Figure 11b), particularly in Northeast China, North Central China, and the Loess Plateau. In some areas, wind erosion showed a negative relationship with the area proportion of cropland, probably due to irrigation by surface water and underground water, which caused the area to be much drier than before. In addition, land-use intensity change from forest or grassland to cropland will increase the range of exposure to bare land, and this conversion often occurs in arid and semiarid areas.

### 4.4. Comprehensive Effects Analysis of the Impact Factors on Wind Erosion

Wind erosion has been attributed to a combination of meteorological factors, human interference, and ecological programs [41,49,52,53]. The driving mechanism of wind erosion is very complex. To more accurately identify the driving factors of wind erosion, we used a pixel-wise multilinear regression model to analyze the effects of different factors. Approximately 17.54% of the TNR of China was affected by the spring average temperature and spring average wind speed, and these areas were mainly distributed in eastern Northwest China. Our results were the same as Jiang et al [35], who found that the wind speed played a leading role in adjusting the wind erosion of Northwest China. North Central China and northern Northwest China were dominated by spring precipitation and spring average wind speed (approximately 9.94%). There were some areas in eastern, northern and central Northwest China and Northeast China that were affected by the spring mean wind speed, covering 7.98% of the TNR of China. Temperature was the main cause of the change in wind erosion in western Northwest China and affected 9.22% of the TNR of China. Our findings are consistent with Li et al [53], who found that temperature played an

important role the wind erosion in the Inner Mongolia. Spring precipitation also influenced the wind erosion of the Loess Plateau and central Northeast China. Overall, approximately 57.58% of the TNR of China was affected by climate change (Figure 12a).

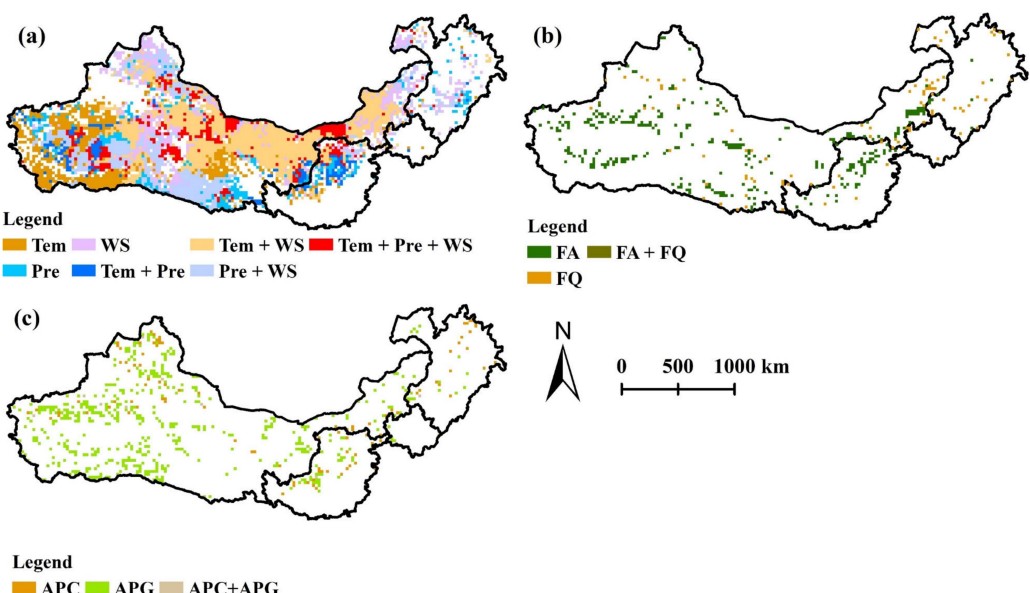

**Figure 12.** Multiple linear regression coefficients of driving factors between wind erosion and (**a**) climate changes, (**b**) ecological programs, and (**c**) human interference. [Temperature (Tem), precipitation (Pre), wind speed (WS), area proportion and quality of forest and shrubland (FA and FQ), area proportion of cropland in a 1km × 1km grid (APC), and area proportion of grassland in a 1km × 1km grid (APG)].

Only approximately 5.93% of the area across the TNR of China was affected by ecological programs; that is, ecological programs improved the ability of local areas to inhibit wind erosion (Figure 12b). Regarding human interference, the change in the area proportion of grassland in a 1 km × 1 km grid influenced approximately 7.30% of the area, which was mainly located in western Northwest China and the northern Loess Plateau. At the regional scale, only a few areas were affected by the area proportion of cropland in a 1km × 1km grid, and the distribution was very fragmented. Human interference affected approximately 8.78% of the wind erosion change in the TNR.

Our study indicated that climate change was the prevailing variable that adjusted wind erosion [42]. Human interference has both reducing and increasing effect on wind erosion [47,54]. Ecological programs, especially for the area proportion of forest and shrubland, also contributed significantly to the dynamics of regional wind erosion [52]. Due to the wide distribution of sandy land and deserts, field observations, meteorological observations, and air observations are required to obtain more accurate information about wind erosion risks. Suitable human activities and ecological programs should be adopted in different locations. For desert areas, planting protective forests of shrubland and grassland to control wind erosion is necessary. To reduce wind erosion in grassland, the recommended steps are to (1) alleviate the impact of grazing by implementing a no-grazing policy in spring and (2) alleviate grazing intensity by reducing the number of livestock. For forest and shrubland areas, ecological programs should be complemented to hold soil. For cropland areas, no-till farming and the stubble left should be effectively executed in the process of agricultural production.

While total wind erosion has been alleviated in the TNR of China, it has not radically changed or worsened in the sandy land and desert cores [16]. The wind erosion intensity is still increasing in parts of the TNR of China, and 15.75% of the study area still suffers from devastating wind erosion. The persistent complement of the ecological programs and

its proper management in different parts of the TNR of China are still required to achieve vegetation recovery and improve the region's ecosystem services.

## 5. Conclusions

Within this study, we used the RWEQ to illustrate the spatiotemporal dynamics of wind erosion in the TNR of China from 1981 to 2020. Furthermore, we quantified the effects of climate changes, ecological programs, and human interference on wind erosion, in the process of determining the driving mechanism of wind erosion. The results showed that China's wind erosion in the TNR showed an obvious decreasing trend from 1981 to 2020. However, the decrease in wind erosion was mainly distributed in Northeast China, while the area of increase was widely distributed in western Northwest China. Climate changes, i.e., the change toward warmer, wetter, and lower wind speeds in northern China, are decisive in reducing wind erosion. However, ecological programs have a limited contribution to the reduction in wind erosion because the area of forest and shrubland increased very little in the vast dry zone. This study depicted the wind erosion dynamic across northern China for a long period, almost 40 years, and identified the wind erosion driving mechanism. Based on the above, we first quantified the effect of ecological programs on wind erosion, so it could be regarded as a guidebook for the government and people to develop reasonable ways to mitigate wind erosion. Our research confirmed that the most important thing was not to plant trees to solve the wind erosion problem in arid and semi-arid areas, but to increase shrubs and grasslands) and to reduce human disturbance (deforestation); such revegetation programs not only reduce wind erosion but also increases carbon sinks.

It's imperative to restore vegetation (increase shrubland and grassland) and reduce human disturbance (deforestation) in the severe wind erosion area in other countries. The study could help people to fulfill the sustainable development goals, reduce carbon emissions and improve carbon sink.

**Supplementary Materials:** The following supporting information can be downloaded at: https://www.mdpi.com/article/10.3390/rs14215322/s1, Figure S1: Location of main deserts, sandy lands, mountains, basins and plains in TNR of China.

**Author Contributions:** Conceptualization, J.W. and X.Z.; methodology, J.W., L.Z. and J.F.; software, J.W.; validation, J.W.; investigation, J.W. and X.Z.; data collection, L.Z., J.F and J.L.; data Processing, J.W. and X.Z.; writing—original draft preparation, J.W.; writing—reviewing and Editing, X.Z.; supervision, X.Z.; funding acquisition, X.Z.; All authors have read and agreed to the published version of the manuscript.

**Funding:** This research was funded by National Key Research and Development Program of China (Grant No. 2020YFA0608103-04) and the National Natural Science Foundation of China (Grant No. 31770758).

**Institutional Review Board Statement:** Not applicable.

**Informed Consent Statement:** Not applicable.

**Data Availability Statement:** The data presented in this study are available on request from the corresponding author.

**Acknowledgments:** We thank the anonymous reviewers and editors for their construction comments, suggestions, and edits to the manuscript.

**Conflicts of Interest:** The authors declare no conflict of interest.

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
