# Peer review of "Effects of Ecological Programs and Other Factors on Soil Wind Erosion between 1981–2020"

_remotesensing, doi:10.3390/rs14215322_

Round 1

Reviewer 1 Report

Manuscript ID- remotesensing-1886975

Title- Effects of the Ecological Programs and other factors on soil wind erosion during 1981-2020

Authors- Jinzhou Wu , Xiao Zheng * , Lanlin Zhao , Junmei Fan , Jinghong Liu

Section- Ecological Remote Sensing

The authors seek to determine the simulated soil wind erosion in the TNR with the RWEQ to analyse the tendency of soil wind erosion and mitigate the effects of the dominating factors on wind erosion. The result of this manuscript is quite impressive and has the potential to be published in remote sensing. However, I have some observations:

·         Authors need to improve the introduction section. Refer to some global studies and their findings on this topic.

·         North-west China has a completely different trend pattern compared to other regions. Which factors are majorly responsible for such a trend?

·         In the conclusion section, you have to mention the implications of your research and how it leaves a footprint in scientific research. Try to incorporate your work into global interest by demonstrating how this research has worldwide importance. It would be interesting for the readers.

Therefore, these are some recommendations that may improve your manuscript quality.

Author Response

Dear Review Expert,

Thank you for the comments and suggestions on our manuscript (MS. No. remotesensing-1886975) entitled “Effects of the Ecological Programs and other factors on soil wind erosion during 1981-2020”. We have revised the manuscript in response to each comment/suggestion carefully. Almost all the suggestions have been accepted and any changes to the manuscript are marked using the "Track Changes" function in the revised manuscript. The reviewers’ major comments and questions are listed in highlight, and the responses are given right below the comments or suggestions. The detailed responses are as follows. We hope that this new version is significantly improved and acceptable for publication.

Yours sincerely,

Xiao Zheng (Corresponding author)

Reviewer 2 Report

The manuscript used the revised wind erosion equation (RWEQ) to simulate the wind erosion variation in northern China from 1981 to 2020. To identify the effects of ecological programs, climate change, and human activities on wind erosion, the manuscript used the multivariate linear regression model to identify its driving mechanism. And the manuscript firstly spatialized the ecological programs and their effects on wind erosion. However, some modifications are necessary before the manuscript can be considered for publication.

General and specific comments:

1. As soil information is one of the key factors in RWEQ, it seems to be better to give a much more detailed description of the soil dataset, including its source and properties.

2. The language needs to be improved. The use of language should be carefully checked. There are many unnecessary descriptions. The deficiency of English usage is significate. English polishing is a must.

3. In general, the logic among different sections is poor. Therefore, sometimes, it’s hard for me to get your true objectives.

4. The heading of Fig.2 needs to mention the place where the sample data were collected. And it may be better to add the number of the sample data into Fig.2b, too.

5. Why don’t you use GDP, representing the economic development state, as one aspect of human activities, just using the population density, the area proportion of cropland and grassland in a 1km×1km grid?

6. The uncertainty is another key issue in the model that simulated the wind erosion change. Here, I can find some uncertainty in some datasets that are directly used from the past studies, but some lack the reference, so I think you should scan your manuscript carefully and add a more detailed reference or uncertainty assessment.

7. As vegetation fraction is one of the most important constituents of the wind erosion model, it’s better to detail describe how you get the vegetation fraction and process the original vegetation dataset.

8. Here, you use the multivariate linear regression model to identify the effects of ecological programs, climate change, and human activities. However, they are on different scales, and how do you process the dataset and comprehensively analyze the wind erosion’s driving mechanism?

9. Research gaps and objectives of the proposed work should be justified before the problems formulation section. This paper includes some useful information and the objective of the study is not well defined. The problem statement is not clear and the objectives are obscure. Furthermore, the paper lacks a very clear and good justification for what is new and innovative about the case or this approach.

10. As vegetation change is dominated by climate change and human disturbance, I am wondering if the ecological programs could be independent driving factors in the wind erosion driving mechanism analysis? So here, you should provide some reasons for your use of these factors.

11. It is imperative to give a more detailed description of the population density is get from the multi-source remote sensing dataset.

12. It may be better to provide a comparison of this manuscript and past studies in the discussion sections.

13. Some of the diagram quality must be improved.

14. All forms (fonts, size, line spacing, and reference), especially the reference, should be carefully checked and revised.

Author Response

(The authors gave the same response as above.)
